# Experimental and Numerical Study on Motion and Resistance Characteristics of the Partial Air Cushion Supported Catamaran

**Shijie Lu** [1], **Jin Zou** [1,*], **Yuangang Zhang** [2] and **Zhiqun Guo** [1,*]

[1] College of Shipbuilding Engineering, Harbin Engineering University, Harbin 150001, China; lushijie@hrbeu.edu.cn

[2] Beijing Institute of Mechanical and Electrical Engineering, Beijing 100074, China; zhangyuangang412@163.com

* Correspondence: zoujing19@163.com (J.Z.); guozhiqun@hrbeu.edu.cn (Z.G.); Tel.: +86-1350-450-8472 (J.Z.); +86-18845142429 (Z.G.)

**Abstract:** The Partial Air Cushion Supported Catamaran (PACSCAT) is an innovative design which combines both the characteristics of hovercraft and catamaran. Further, it provides a high-speed and efficient solution with excellent performance, particularly for shallow water. In this paper, experimental and numerical method are carried out for research of motion attitude and resistance characteristics, which provide a reference for further research and hull optimization work. By model towing test and data interpretation, and the resistance, trim, and heave varying law with increasing speed is summarized. From the view of total resistance, the impacts of the cushion pressure and air flow on resistance performance of PACSCAT are analyzed. Based on the theory of viscous fluid mechanics, a numerical simulation method with high prediction accuracy is established. The flow field around and inside the hull is simulated, the simulating results show good agreements with the testing data. Finally, the effect of the cushion compartment improving the resistance performance is studied. The results show that the cushion compartment is significant for adjusting the pressure distribution of the air cushion. And the average resistance reduction ratio at the high-speed segment can even reach 22%.

**Keywords:** PACSCAT; experimental and numerical study; cushion compartment; resistance reduction

## 1. Introduction

The Partial Air Cushion Supported Catamaran (PACSCAT) is an innovative high-performance composite craft based on catamaran hull, assisted by an air cushion between the demihulls. According to Table 1, air cushion assisted vessels are generally classified into four categories by the weight ratio supported by the air cushion. In this study, the proportions of air cushion supported weight is around 25%. The rest of weight is supported by hydrodynamic lift and hydrostatic buoyancy. Forces acting on PACSCAT include buoyancy and cushion lift. While the airflow is injected into the air hover chamber from the plenum chamber, the hull is lifted, and the water in air hover chamber is pressed out by the high-pressure air cushion. This also decreases the wetted hull surface. In 2009, and 2013, a 30 m long PACSCAT by Independent Maritime Assessment Associates (IMAA), and 12 m long technical demonstrator by Harbin Engineering University Ship Equipment and Technology Company (HEUShip, Harbin, China) were launched, respectively. Generally speaking, the motion and resistance characteristics of the PACSCAT is affected by the interaction of cushion aerodynamics and hydrodynamics of the hull. However, plenty of papers published in public are concerned with the performance of the air cushion vehicle (ACV) and surface effect ship (SES), which ignored the

sub-variety of the PACSCAT. Casey carried out two different air flow plenum schemes on the model of T-Craft (a ACCAT design). The relationship between lift weight ratio and drag growth generated by bow seal was discussed [1]. A large number of SES-100A/B test results were compared and analyzed with real ships' data. Wilson suggested that the series of SES-100A/B can ignore the wave-making resistance, but it affects the wetted surface significantly [2]. In addition, the air cushion compartment is introduced in ACV and SES research. This technique has a positive effect, but no PACSCAT form has been studied. Study on the drag reduction effect of the air cushion compartment of an SES ship is carried by towing test.

**Table 1.** Categories of air cushion assisted vessels by air cushion support ratio.

| Category | Air Cushion Vehicle (ACV) | Sidewall Hovercraft | | Partial Air Cushion Supported Catamaran (PACSCAT) |
| --- | --- | --- | --- | --- |
| | | Surface Effect Ship (SES) | Air Cushion Catamaran (ACCAT) | |
| Air Cushion Support Ratio | 100% | 100%~75% | 75%~50% | 50%~20% |

However, the results showed that the total resistance of the sidewall hovercraft is increased because the separator seal generated a high seal-resistance [3,4]. Doctors took a model test on a sidewall hovercraft Scot09, with a thin plate demihull. The results showed a positive effect of the air cushion compartment [5]. Neu studied the resistance performance of T-craft and achieved a good prediction accuracy [6]. Maki calculated the resistance of the SES in still water. The results were in good agreement with Doctors' results [7]. Yang analyzed the variation law of resistance with the changing airflow rate based on a PACSCAT towing test data [8,9], but no further analysis on the relationship between the cushion pressure distribution and motion characteristics was performed. Guo introduced a seakeeping analysis method for the PACSCAT by combining the 2.5-D theory and simplified wave-equation [10], but only seakeeping performance is discussed.

The researches mentioned above cover multiple aspects of air cushion assisted vessels research work, including the air hover system, seakeeping performance, resistance, and other characteristics. But relatively few researches have been conducted focusing on the acting mechanism of air cushion pressure and motion characteristics of the PACSCAT, which is directly related to the resistance performance and other motion characteristics. Therefore, in this paper, the aim is focused on the acting mechanism of air cushion pressure and motion attitude of the PACSCAT by combing an experimental and numerical method. Moreover, for another objective, the air cushion compartment method is applied to the origin model to verify the acting mechanism of the air cushion and the resistance reduction effect together.

The following sections are organized as: first, a brief description of the hull and the towing test arrangement are shown. Results and analysis of the experiment (including pressure, sinkage, trim and resistance) are shown in detail. Second, the numerical method is developed, and the simulation results are verified and analyzed by comparing with the experimental data. Finally, air cushion compartment models are introduced, and the resistance reduction effect is discussed based on numerical analysis.

## 2. Model Towing Test and Numerical Setup

### 2.1. Geometrical Description of the PACSCAT Model

The main configuration and geometric features of the Partial Air Cushion Supported Catamaran Model is shown, while the transverse section profile line is also attached in Figure 1. The main parameters are shown in Figure 2 and Table 2. It can be observed that the main body of the ship is composed of two firm demihulls connected by a connection bridge structure. An air-pressure plenum chamber sits in the middle of the connection bridge bottom, and the air cushion chamber sits right below the connection bridge linked to the plenum chamber, which is bounded by rigid demihulls

and flexible air seals at the bow and stern. The air cushion mainly exists in the air cushion chamber, where the movement of the hull, the effusive air flow at seal bottom, and plenum fan meet a dynamic balance. The dynamic procedure maintains stable cushion pressure.

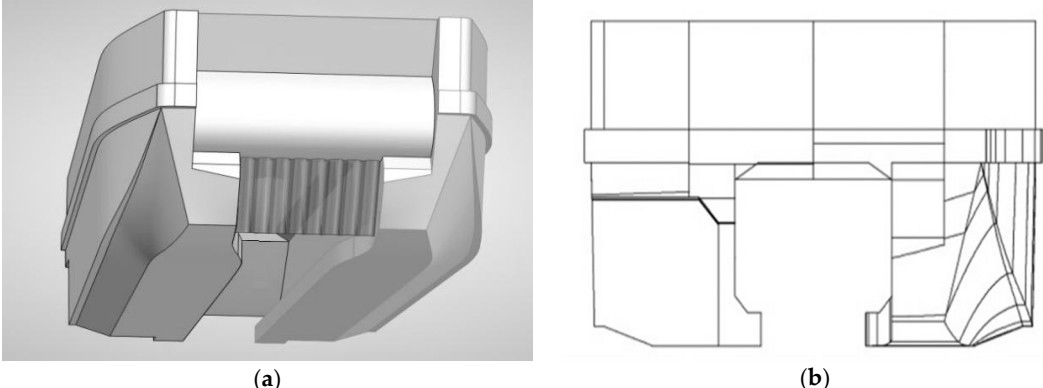

(**a**)                                        (**b**)

**Figure 1.** Model in towing test. (**a**) 3-D model of the Partial Air Cushion Supported Catamaran Model (PACSCAT); (**b**) Lines plan of the PACSCAT hull.

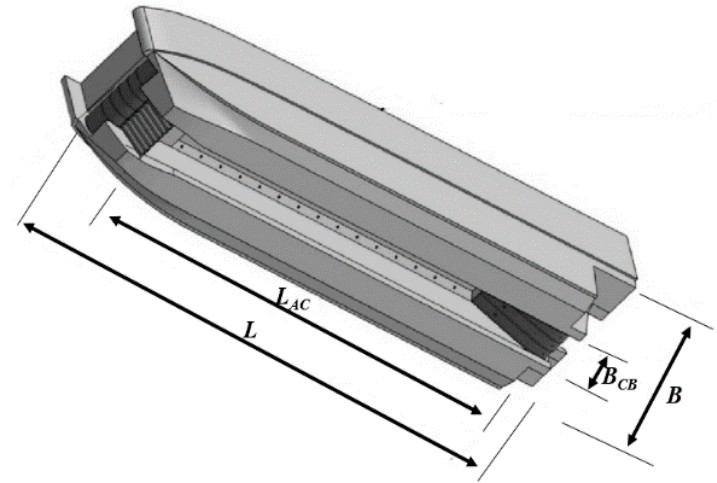

**Figure 2.** Main dimensions of the hull.

**Table 2.** Main dimensions.

| Main Feature | Symbol | Value |
|---|---|---|
| Length overall (m) | L | 3.0 |
| Beam overall (m) | B | 0.70 |
| Connection bridge Beam (m) | $B_{CB}$ | 0.26 |
| Connection bridge height (m) | $H_{CB}$ | 0.11 |
| Air cushion chamber length (m) | $L_{AC}$ | 2.4 |
| Demihull beam (m) | $B_D$ | 0.22 |
| Longitudinal center of gravity (from rear) (m) | $L_{CG}$ | 1.42 |
| Designed displacement (kg) | $\Delta D$ | 145 |
| Light displacement (kg) | $\Delta L$ | 90 |
| Full-load displacement (kg) | $\Delta F$ | 160 |
| Calm water forward draft(m) | $T_F$ | 0.128 |
| Calm water aft draft (m) | $T_A$ | 0.154 |

## 2.2. Experimental Setup

The PACSCAT model tank test was carried out in the China Special Vehicle Research Institute (also known as Aviation Industry of China NO. 605 subsidiary research institute) with the maximum length

towing tank in China. The towing tank principal dimensions are: length 510 m, width 6.5 m, and depth 6.8 m. The corollary carriage towing system can reach a speed range of 0.1 m/s to 22 m/s, with a stable speed error under 0.1%. A schematic overview of the tank test is shown in Figure 3. The PACSCAT model is fixed to the carriage (with pitch and heave free). The arrangement of measuring device and sensors are as follows:

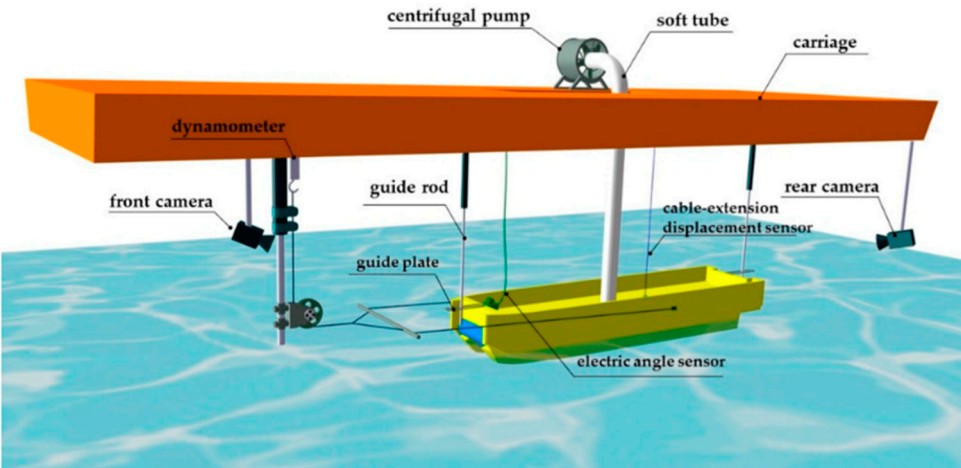

**Figure 3.** View of the experimental setup.

During the towing tests, a dynamometer was attached to the carriage to measure the resistance (The longitudinal position of the towing point was at the position of the gravity center, while the vertical position should also be located to the gravity center as close as possible); and an angle sensor was mounted at the foredeck to measure the trim angle; while a cable extension displacement transducer was mounted at the gravity center to measure the sinkage. In the hover air chamber, the air cushion pressure along the longitudinal direction was measured by 5 isometric pressure sensor probes, which were mounted along one of the inner sides of the demihull. In addition, cameras mounted before and after the hull were used to monitor the wave-making characteristics. As for the inner hover air chamber, a camera was also used for wave-making monitoring and jacklights were introduced for a clear view-record. Auxiliary gridlines were introduced to measure inner wave-making and draft, details can be found in Figure 4.

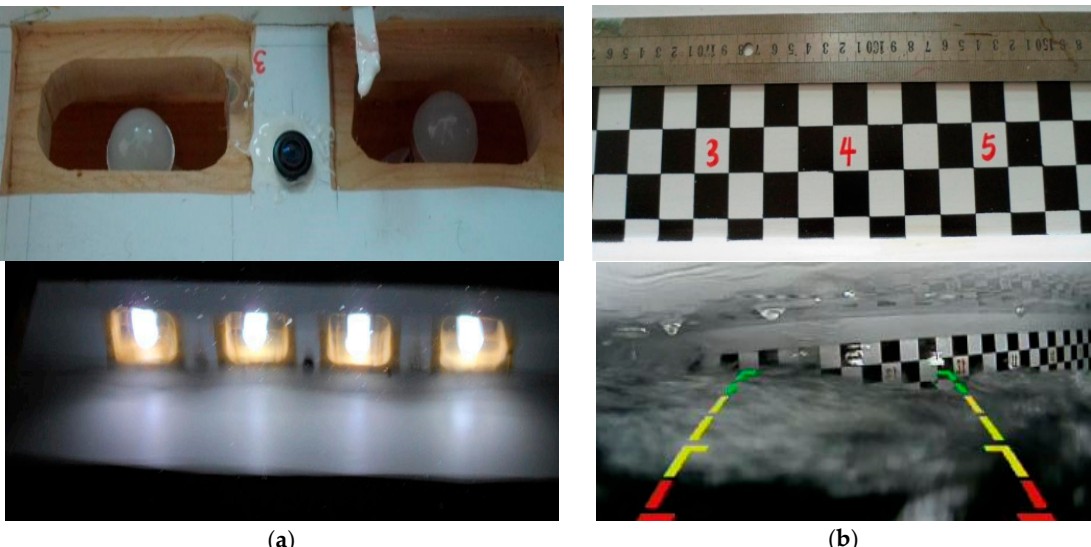

| (a) | (b) |

**Figure 4.** Inner cushion chambers devices. (**a**) Inner camera and jacklights; (**b**) Inner gridline and working condition.

The internal flow passage of the PACSCAT hover system is as shown in Figure 5. An electric powered centrifugal fan was mounted to generate the air flow, and the air, through a segment of soft tube, was injected into the plenum chamber. The air flow was stabilized in the plenum chamber. And though the uniform arranged air inlet holes, the flow was led into air cushion chamber. So was the bow and stern seal. To keep a stable quantity of air flow, a gas turbine meter was attached to the hover flow system, as introduced in paper [9] by Yang J.L.

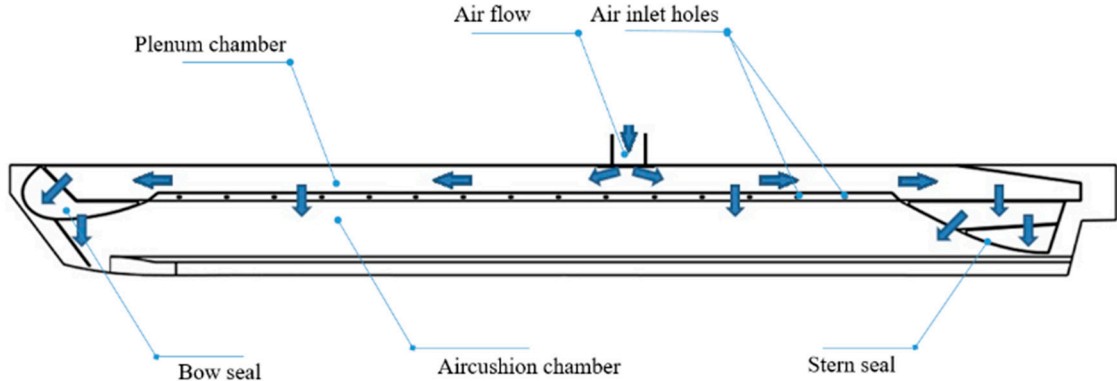

**Figure 5.** Internal flow passage of the PACSCAT hover system.

### 2.3. Experimental Conditions and Data Processing Method

In this paper, a series of experiments were carried out—parameters varies from displacement, longitudinal center location, and air flow rate—the trail tested three different displacement situations: 90 kg, 145 kg, 160 kg, at two longitudinal center location of gravity: Xgm = −150 mm, Xgm = −80 mm, under 5 different air flow rate: Q = 220 m³/h, Q = 180 m³/h, Q = 150 m³/h, Q = 125 m³/h, and Q = 80 m³/h. Therefore, under different working conditions, the air cushion pressure, heave, pitch, and total resistance change with speed have been obtained. According to the camera records, the hull wave-making, motion attitude, and wave inside the hover air chamber could be observed. In addition, air cushion pressure along the air cushion was measured by the pressure sensor. The information is advantageous to the detailed analysis of the trial results.

In this paper, data dimensionless processing was carried out as: dimensionless air cushion pressure ($P_c/\rho gh$), the drag-to-weight ratio ($R/\Delta$), and dimensionless sinkage ($h/L$). All these parameters versus the volume Froude number are summarized with figures. In the following Sections 3.2 and 3.3, further experimental phenomena and results analysis will be discussed.

### 2.4. Mathematical and Numerical Models

According to the experiment, the PACSCAT shows different characteristics in each speed segment. To further simulate and study the attitude and resistance performance during the whole-speed segment, for which the numerical simulations were carried out for Froude numbers ranging from 0.09 to 1.1. The velocity distribution was consistent with the trial conditions.

To establish a comprehensive solving system for the hull surrounding incompressible viscous flow field, software CFX based on FVM was adopted in this paper to solve the flow field around the hull. The governing equations include continuity equations, momentum conservation equations, and Navier–Stokes equations. The system is closed by introducing the SST (Shear Stress Transfer)k-ω turbulence model. And the VOF (Volume of Fluid) method is introduced to deal with the free surface deformation model. In the trial, heave and trim angle were measured to extract the navigation attitude. Therefore, motion equations force and moment equilibrium were introduced, the 2-DOF motion is simulated by the following dynamic solver procedure:

$$\vec{F} = \mathrm{m}\frac{d^2\vec{X}}{dt^2} \tag{1}$$

$$\vec{M} = \frac{d}{dt}\left(I\frac{d\vec{\theta}}{dt}\right) \tag{2}$$

where $I$ presents the hull gravity center inertia mass matrix.

First, based on initial meshes, the continuity equation and Navier–Stokes equation system are solved. Then, the fields of shear stress and pressure around hull-surface are integrated to calculate the force and moment acting on the model. The equations which integrate the shear stress and pressure are described as follow:

$$\vec{F} = \int_S ([\tau] - p[I]) \bullet \vec{n} dS - \vec{G} \tag{3}$$

$$\vec{M} = \int_S \left(\vec{r} - \vec{r}_G\right) \times \left(\vec{\tau} - p[I]\right) \cdot \vec{n} dS \tag{4}$$

where $[\tau], p[I]$, and $\vec{G}$ are shear stress, pressure, and gravity, separately. While $S$ is symbol for the surface of hull and the $\vec{r}$ and $\vec{r}_G$ for displacement of mesh nodes and gravity center. After that, the linear and angular motion parameters, obtained by solving Equations (1) and (2), are used to update the hull position. The same applies to the mesh nodes. Subsequently, a loop is carried out to solve the fluid field with the new mesh. This algorithm meets the termination criteria at the phase where the force and moment stabilize at a certain value. Figure 6 shows the solving procedure in detail.

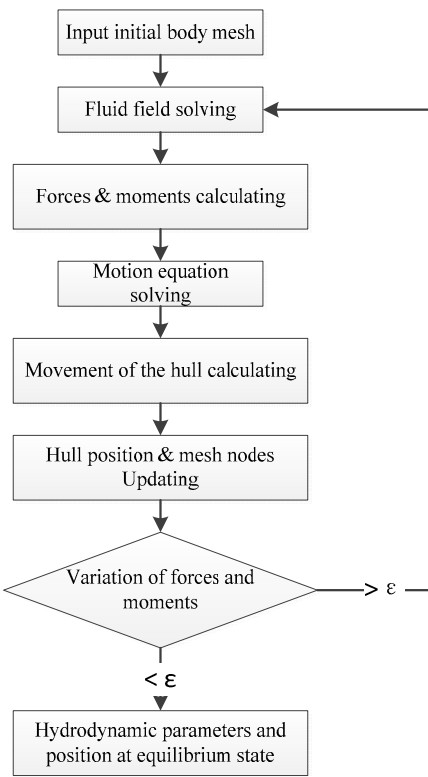

**Figure 6.** Solving procedure of the algorithm.

*2.5. Boundary Conditions and Mesh Generation*

To simulate the flow field around the PACSCAT, the primary concern is the establishment of a calculating domain. As shown in Figure 7, due to the symmetry principle, a symmetry model and domain was introduced. Considering the Froude ranges from 0.09 to 1.1, the dimensions have a significant impact on the accuracy of hull wave-making in the stern flow field. Hence, the total length of the domain was about 6 L. Where the stern of the model was taken as the origin of the X coordinate. The domain extended 2 L from the front of the model, and 3 L from the rear. The width of the domain

extended 1.5 L. Water depth under the free surface was 2.5 L, and air extended 1 L above. The initial attitude of the hull was adjusted according to the model attitude obtained in the model tank trial.

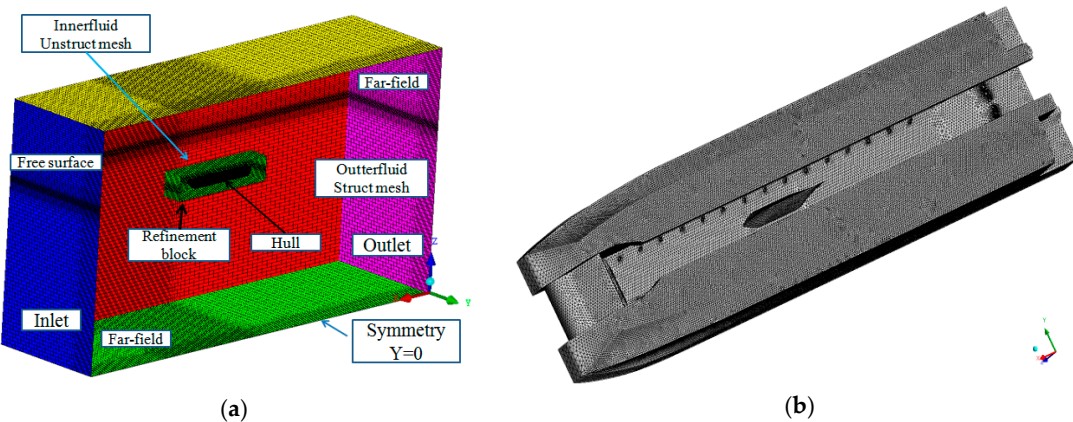

**Figure 7.** Calculating mesh. (**a**) Calculation domain setup and set boundary conditions; (**b**) Hull mesh.

The boundary conditions are specified as follows: The calculation domain top, bottom, and side planes are considered as free-slip walls. The inlet, domain plane in front of bow is imposed as the fluid velocity inlet, where the velocity is set as the trail drag velocities. Oppositely, the opposing plane is considered as the pressure outlet. The hull surface is considered as a rigid and a no-slip wall boundary, and the longitudinal center plane is specified as a symmetry plane. At the top of the model's air stabilize chamber a velocity inlet is added, which the volume fraction considers as 1, namely, simulating air inlet fan. The water-immersed seals' resistance is the main portion of the seals' resistance, in this paper, Seals are regarded as a rigid planing surface because it can maintain a relatively rigid state after the force balance under the interaction of the incoming flow and air flow.

To improve work efficiency and computational accuracy, the calculation domain is divided into two parts, Innerfluid region and Outerfluid region. Respectively, the outerfluid region was filled by relatively sparse hexahedron mesh, the innerfluid region was filled up with tetrahedral mesh to adapt to the complex hull surface. And mesh was densified with a growth rate at the free surface. Dimension 0.003 L was used to in the hull surface grid generation, and a layer mesh (10 layers of prism mesh at ship bottom and side, five layers of prism mesh in side hover system) was attached to the hull surface. With an initial y+ of 60, and according to Equation (5), set the first boundary-layer thickness $\Delta y_p$ with growth rate 1.2. Then the automatic near-wall treatment is adopted.

$$y+ = 0.172\left(\frac{\Delta y_p}{L}\right)\mathrm{Re}^{0.9} \tag{5}$$

where, $\Delta y_p$ represent for first boundary-layer thickness. The overall grids number is about 2.8 million. During the transient computing course, each physical time step is set with 10 internal iterations. And the time step is given with the CFL (Courant Friedrichs Lewy) Equation [11]. A CFL, also abbreviated as Courant number, shows the dimensionless relationship among velocity U, local mesh size $\Delta x$, and time step $\Delta t$.

$$CFL = \frac{U\Delta t}{\Delta x} \tag{6}$$

## 3. Experimental and Numerical Results

In this section, experimental and numerical results data are extracted and analyzed. The relationship among parameters, such as air cushion pressure, trim angle, heave, and resistance is preliminarily analyzed. In addition, the components of total resistance is divided.

## 3.1. Results of Towing Test

Main parameters dimensionless have been proceed under dimensionless expressions shown in Table 3. In Figures 8–10, it can be seen that with the change of load displacement (90 kg, 145 kg, 160 kg), the variations of air cushion pressure $P_c$ are consistent with each other. At light displacement 90 kg, with the increase of air flow, the air cushion pressure increased. Within the range of *Fr* = 0.59–0.66, the pressure curve reached its pressure peak, which was slightly later than the resistance peak (*Fr* = 0.46–0.66).

Within the range of the whole-speed segment, interaction of the air cushion pressure, and hydrodynamic force act on the model. Air cushion pressure had a small influence on the trim angle, but a great relationship with the heave. In the low-speed segment (*Fr* < 0.46), with the increase of speed, the air cushion pressure was stable within a certain range while the trim angle increased, thereupon, the center of gravity position decreased, and the value of heave went down. In this speed segment, the hull met the characteristics of conventional displacement ship, and the resistance increased sharply.

Within the resistance peak segment (*Fr* = 0.46–0.66), the cushion pressure gradually increased, the heave height increased, and the trim angle also increases continuously. Under the combined action of these three factors, the total resistance of the ship increased sharply up to the peak.

**Table 3.** Main parameters and dimensionless expression.

| Main Parameters | Symbol | Dimensionless |
| --- | --- | --- |
| Cushion pressure | $P_c$ | Pc/ρgh |
| Heave | h | h/L |
| Trim | deg | deg/° |
| Resistance | R | R/Δ |

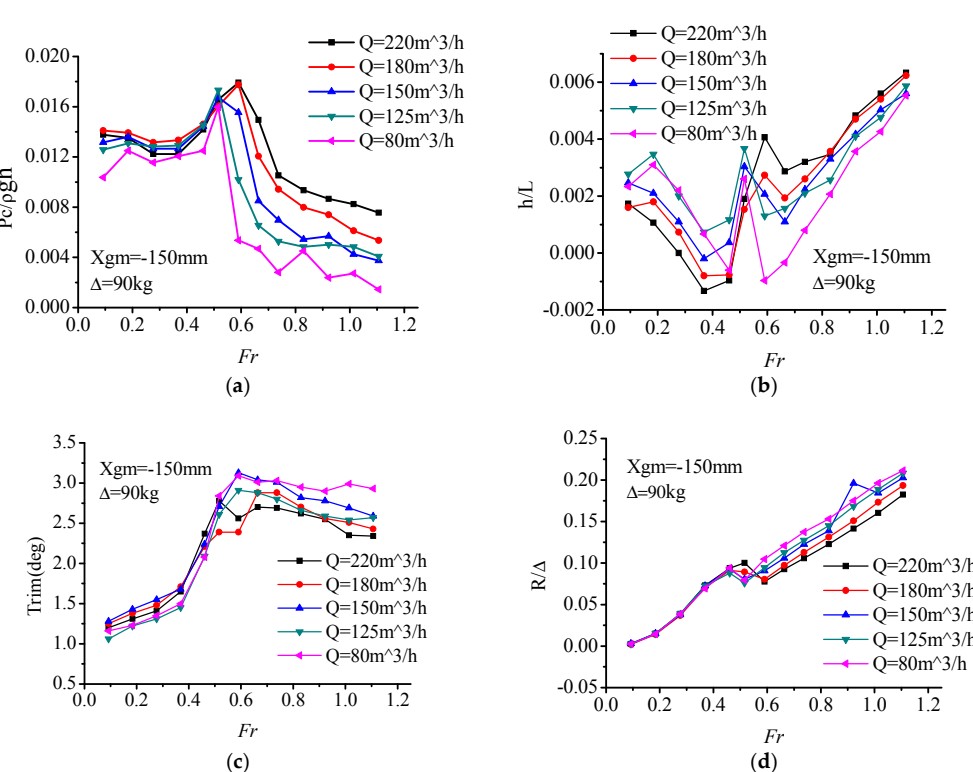

**Figure 8.** Comparisons of numerical and experimental results with consistent displacement Δ = 90 kg. (**a**) Curves of air cushion pressure; (**b**) Curves of air cushion heave; (**c**) Curves of trim angle; (**d**) Curves of total resistance.

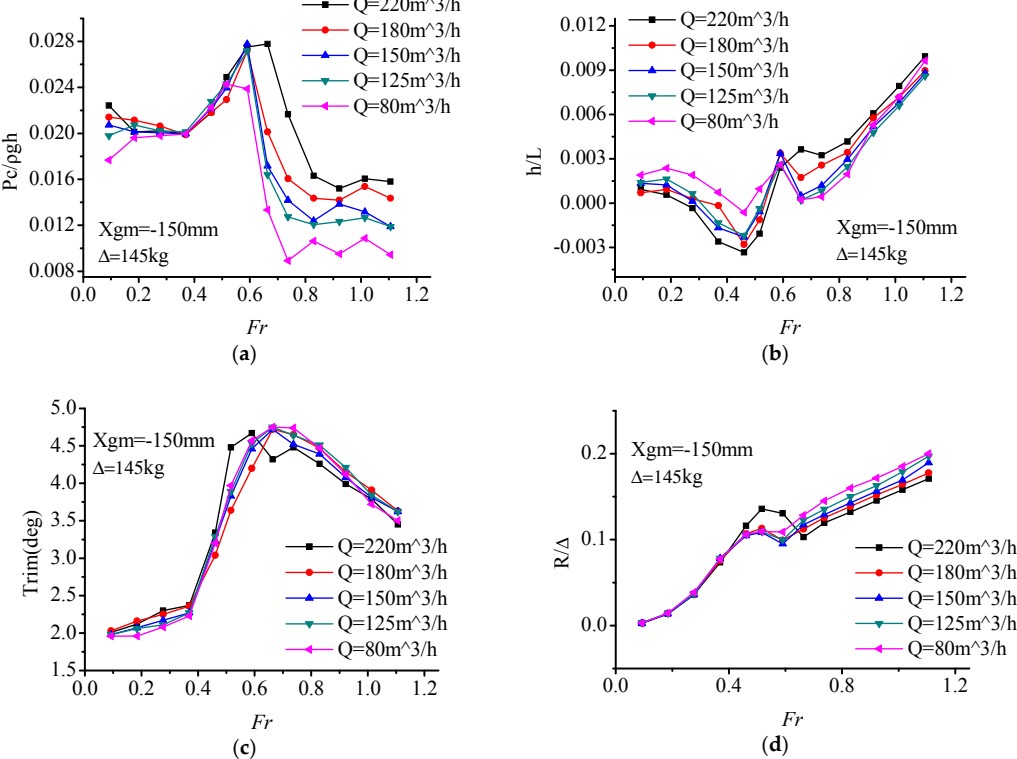

**Figure 9.** Comparisons of numerical and experimental results with consistent displacement Δ = 145 kg. (**a**) Curves of air cushion pressure; (**b**) Curves of air cushion heave; (**c**) Curves of trim angle; (**d**) Curves of total resistance.

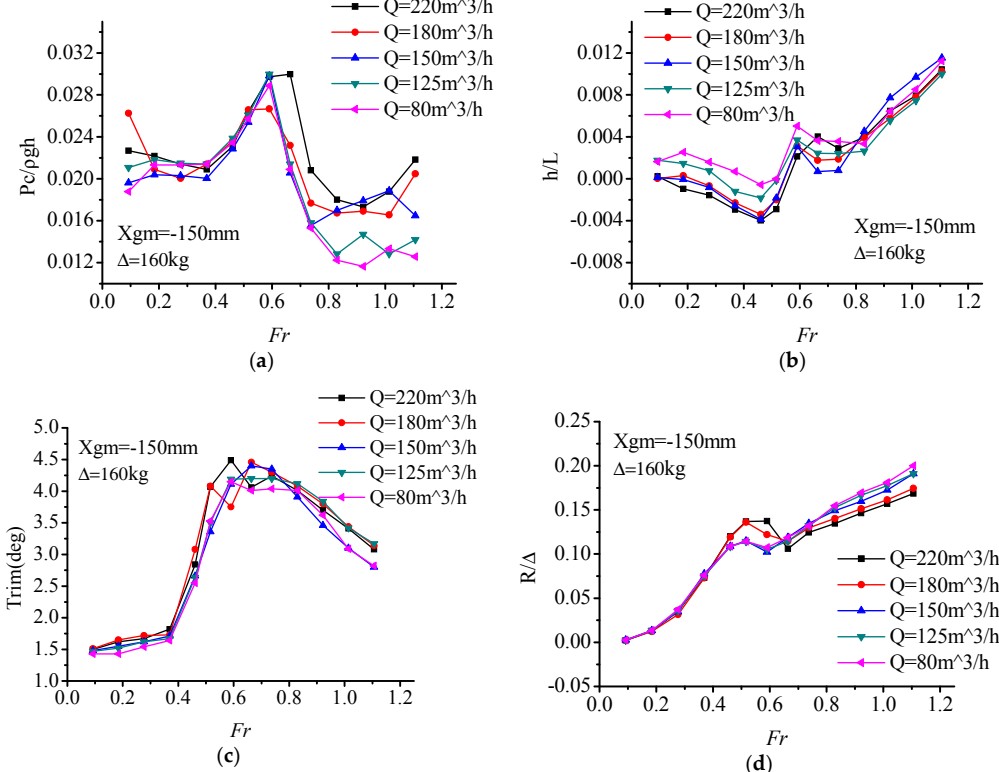

**Figure 10.** Comparisons of numerical and experimental results with consistent displacement Δ = 160 kg. (**a**) Curves of air cushion pressure; (**b**) Curves of air cushion heave; (**c**) Curves of trim angle; (**d**) Curves of total resistance.

In the high-speed segment ($Fr > 0.66$), the air cushion pressure reached a peak, so did the heave height. The trim maintained at a high angle level with a slightly reducing trend. Furthermore, on account of the high value of trim angle, the bow-flow-discharge increased, and air cushion pressure also reduced gradually, which caused the air cushion lift to have a decreasing trend. Total resistance increased gradually due to catamaran hull high-speed planing hydrodynamic factors.

To sum up, during the low and high-speed segment, the influence of cushion pressure on the PACSCAT sailing state is not obvious, and the response pattern is consistent with the hydrodynamic influence trend. While in the range of the resistance peak, cushion pressure affects the resistance peak by changing the motion attitude. This phenomenon may be caused by the interaction of cushion pressure and the motion state. At lower speed, the cushion influence is limited. The total resistance is similar to a displacement hull. While the cushion pressure and trim make the heave decrease gradually. But in the resistance peak-speed segment, the cushion chamber remains at a high-pressure level which generates a high lift ratio and moment, the uppitch degree increases rapidly. At the same time, the high lift ratio causes an increase of the heave. At the high-speed segment, the influence of the cushion pressure weakens; the response pattern is consistent with the hydrodynamic influence trend again.

As for the total resistance of PACSCAT, the influence of the air flow supply rate in the low-speed segment is finite, with a narrow resistance difference between each condition. In the resistance peak segment, the large air flow rate causes a large resistance peak. However, in the high-speed segment, the larger air flow rate matches a smaller resistance. That means the whole-speed segment's optimal resistance curve can only be obtained by selecting suitable air flow rate schemes for specific speeds, respectively.

In another form, the experimental data are organized to clarify the comparison of the air cushion pressure, heave, trim, and resistance under various displacements at the same air flow rate, respectively (On account of five different air flow rate condition, the figures are summarized in five groups, and the number plot is up to 20. Considering a better continuity and readability of the main text, the plots are attached in Appendix A). From Figures A1–A5, it can be seen that the demihull draft increases with the displacement, so does the trend of air cushion pressure and trim angle. As for the total resistance, the displacement has a significant influence on total resistance peak value, with a positive correlation trend; the increase of displacement corresponds to the increase of the resistance peak while there is no obvious resistance difference at high speed.

### 3.2. Test Phenomena and Resistance Components Analysis

Due to the resistance, components of PACSCAT are complex, both the wider demihull and air cushion have a significant influence on the sailing performance, which distinguishes it from traditional hovercraft. Using the estimation formula to extract air cushion and hull resistance (air cushion wave-making resistance, air seal resistance, air resistance, friction resistance, and so on), respectively, the composition and proportion of the resistance along the whole-speed segment were obtained.

The air cushion wave-making resistance of a hovercraft can be expressed as:

$$\frac{R_{aw}}{\Delta} = \frac{C_w^N Pc}{\rho_w g L_c} = \frac{\rho_a}{\rho} C_w^N \overline{p}_c \tag{7}$$

where $C_w^N$ represents Newman cushion wave resistance coefficient valuing from the resistant coefficients atlas [12].

The total resistance generated by the hull can be expressed as:

$$R_{hull} = R_{hw} + R_{air} + R_f + R_{seal} \tag{8}$$

where, $R_{hw}$, $R_{air}$, $R_f$, and $R_{seal}$ represent hull wave-making resistance, air friction, friction, and total seal resistance of bow and stern, respectively.

Further, hull wave-making resistance and air friction can be obtained by

$$R_{air} = \frac{1}{2}\rho_{air}C_d V^2 A_T \tag{9}$$

$$R_f = \frac{1}{2}C_f\rho V^2 A_w \tag{10}$$

where $\rho_{air}$: air density, $C_d$: coefficient of air resistance, $V$: speed, $A_T$: model maximum cross-sectional area. $\rho$: water density, $C_f$: frictional resistance coefficient, adopting the 1957 ITTC (The International Towing Tank Conference) formula, $A_w$: wetted surface.

And, air seal resistance can be calculated by formula [13]

$$R_{bowseal} = (P_{aft} - P_{fwd})B(l_2 - \frac{l_2^2}{(l_2+l_1)})\sin\alpha +$$
$$\frac{0.075}{(\lg(\frac{V(l_2-l_1)}{v})-2)^2} \times \frac{1}{2}\rho V^2 B(l_2 - l_1)\cos\alpha \tag{11}$$

$$R_{stemseal} = \frac{0.075}{(\lg(\frac{Vl_{wet}}{v})-2)^2} \times \frac{1}{2}\rho V^2 Bl_{wet} \tag{12}$$

where, $P_{aft}$: rear side pressure of seal, $P_{fwd}$: front side pressure of seal, $B$: seal width, $l_2$: wetted length of the seal, $l_1$: non-wetted length of the seal. The force diagram sketch is shown in Figure 11.

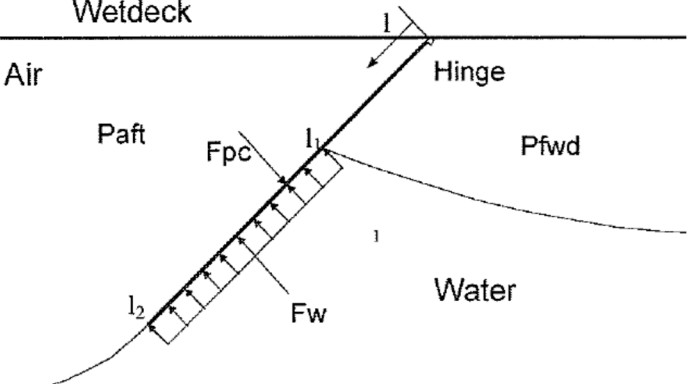

**Figure 11.** Force diagram sketch of the seal.

By calculating the difference between Rt, (total resistance) and other resistance components, total wave-making resistance Rw (including air cushion and hull wave-making resistance) is obtained. In the actual situation, both $R_{aw}$ and $R_{hw}$ are interacted and mixed. Using $R_w$ is more likely to explain the composition and proportion of the wave-making resistance.

Through data processing, it is worth noting that the test phenomena and resistance characteristic vary with the speed and have strong regularity and have basically the same trend under various working conditions. Taking condition $\Delta = 145$ kg, Q = 150 m³/h as a representative sample is feasible. Decomposing the resistance, every component force and its ratio are obtained within the range of speed. Some of the components (e.g., air resistant) have a very small ratio and are omitted. Figure 12 shows a diagram of the forces ratio situation. And Figure 13 shows a series of wave snapshots of bow view at different speeds.

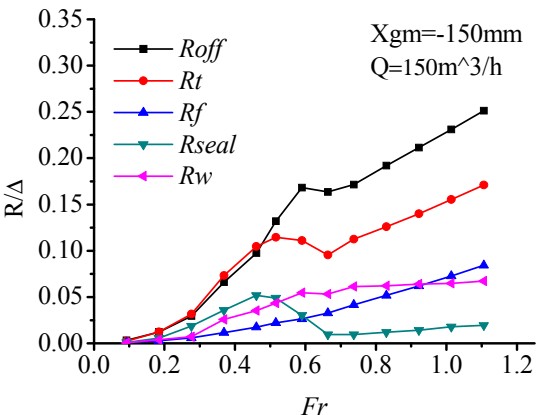

**Figure 12.** Resistance components Comparison under condition Δ = 145 kg, Q = 150 m³/h.

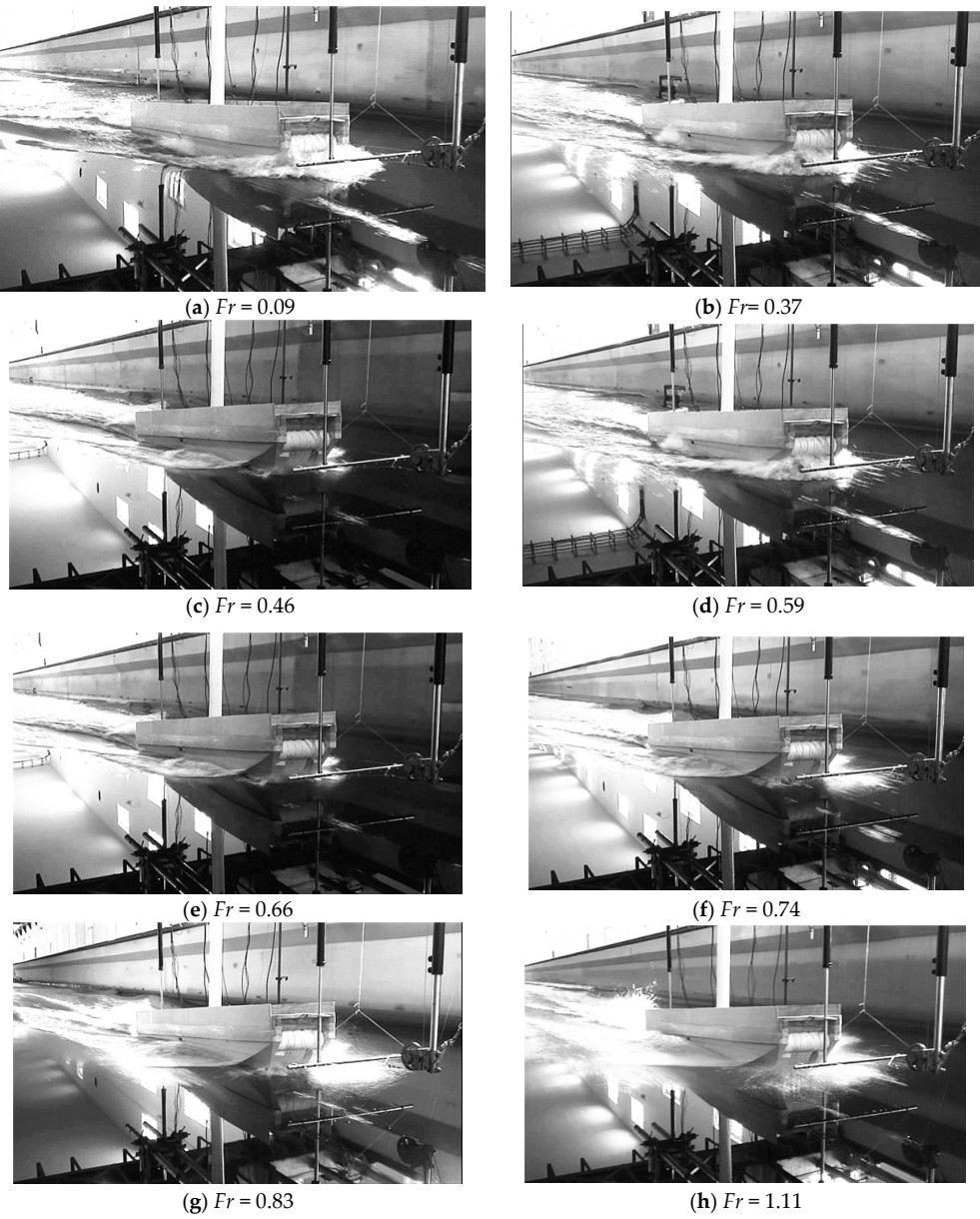

(**a**) *Fr* = 0.09

(**b**) *Fr* = 0.37

(**c**) *Fr* = 0.46

(**d**) *Fr* = 0.59

(**e**) *Fr* = 0.66

(**f**) *Fr* = 0.74

(**g**) *Fr* = 0.83

(**h**) *Fr* = 1.11

**Figure 13.** Snapshots of the planing trimaran at different speeds. Δ = 145 kg, Q = 150 m³/h.

Where $R_{off}$ is total resistance of hover system off, $R_t$ is total resistance with hover state, $R_f$ is frictional resistance dynamically calculated with a wet surface, $R_{seal}$ is total resistance of bow and stern seal, $R_w$ is wave-making resistance (including air cushion and hull wave-making resistance).

From Figure 13, total resistance comparison between $R_{off}$ and $R_t$ at the low-speed segment ($Fr < 0.46$), $R_{off}$ is slightly less than $R_t$. The cause is that the bow seal submerged depth is great at the low-speed hover state, and the impact of decreased hull wetted area cannot cover the friction resistance and wave-making resistance produced by bow air seal. Similarly, the total seal resistance has an increasing trend because of the heave and trim making seal submerged depth increase. At the Froude number of 0.46, the seal immersed maximally, the total seal resistance reached the top until the Froude number of 0.66 where the seal reached the free surface gradually, the total seal resistance decreased to a negligible level. The frictional resistance was similar to that of the displacement ship which increased monotonically with the increase of speed. At low speed, the proportion of friction resistance was small, but with a major proportion at high speed. The wave-making resistance takes a large proportion of total resistance in the full-speed range. And specifically, it is the maximum component at resistance hump, and getting to a stability range after that.

In low-speed segment ($Fr \leq 0.37$), with the speed increasing the model trim raised slightly and the gravity center descended obviously, which deepens the immersed depth of the seal. High-speed airflow leaks from the seal slit and its junction with the hull. The water-pushing phenomenon of the bow air seal is obvious. At this moment the seal resistance takes a huge proportion of the total resistance. In resistance peak exceed segment ($0.37 \leq Fr \leq 0.59$), the wave-making length was approximately equal to the length of the air cushion. The bow and stern seals were at the wave crest. The bow seal water-pushing phenomenon was intensified. With the heave and trim change, at the front of demihull, the wave-making was also severe. The wave dashes on the bow seal. With all factors above coupling, the seal resistance is high. During this segment, the wave-making trough in hover chamber was located at midship, it increased the side overflow of the air partly. However, the overflow can be omitted under high displacement working condition. When the Froude number was over 0.59 ($Fr > 0.59$), the wave-making trough in hover chamber moved to stern seal, and cushion pressure and trim angle meet the maximum, and also explains that with speed increases, the model was in a relatively stable state, with the bow the seal lifted out of the water, hence, the air seal resistance reduced to the minimum. The seal also makes airflow leaking height increase which caused the cushion chamber pressure to decrease obviously. It also caused waves at the bow chine line and stern, generating spatter and breaking waves and energy dissipated.

### 3.3. Validation and Results of Numerical Method

To evaluate the results of the numerical simulation method, comparisons between the wave phenomenon of numerical simulation and experimental results were carried out. So were the numerical simulation and experimental parameter values. The numerical method showed an accurate simulation of these characteristics with better anastomosis. Taking conditions $Fr = 0.31$ and $Fr = 1.03$ as examples, details are shown in Figures 14 and 15. At the Froude number of 0.31, the gravity center and the trim angle were both a small value. Bow seal generated a water-pushing effect and wave at midship side reached a certain height. At the Froude number of 1.03, no significant bow appeared, and the wave peak moved to the stern. A larger wake flow crests formed at the longitudinal mid-section which were all reflected in numerical simulation. In experiment, there were some thin non-airtight slit on the seals and a slit may also exit at the junction with the hull while in the numerical situation, all these junction and slit could be ignored, only demihull side leaks air flow as shown in Figure 16. Therefore, high-speed airflow can leak from the slits which may cause the wave-making, especially wave-making behind the hull, to be more obvious than the simulating condition. Figures 14 and 15 reflect the phenomenon. It is also a reason explaining why the experimental resistance is higher than the numerical result.

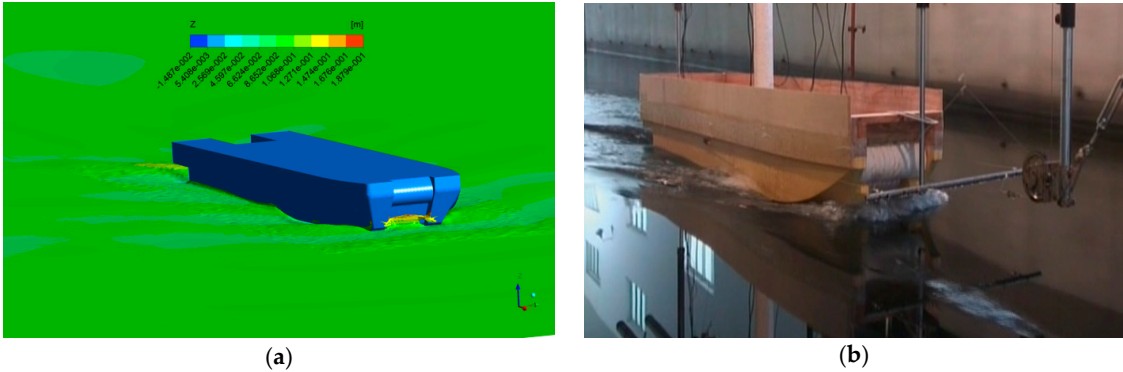

|  (a)  |  (b)  |

**Figure 14.** Comparison of numerical and experiment free surface wave making (*Fr* = 0.31).

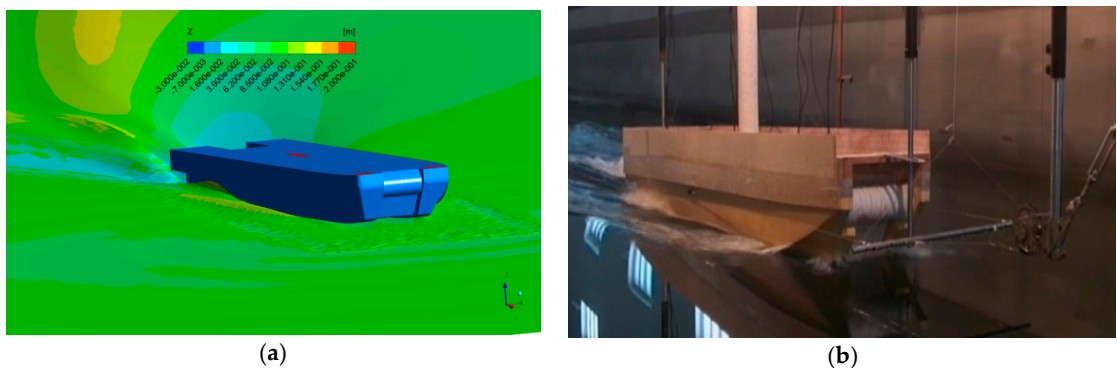

|  (a)  |  (b)  |

**Figure 15.** Comparison of numerical and experiment free surface wave making (*Fr* = 1.03).

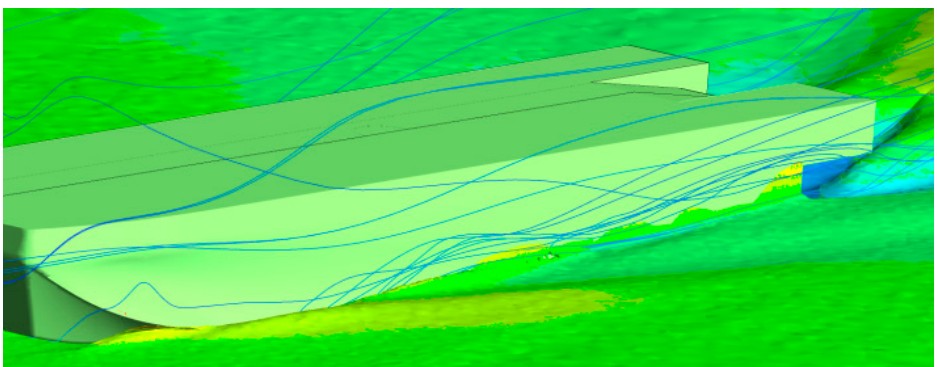

**Figure 16.** Air leaks from the demihull side.

Table 4 shows the comparison of the numerical calculation and model test results (Δ = 90 kg, Xgm = 150 mm), and calculated R/Δ, trim angle, sinkage, and numerical error are summarized in Figures 17 and 18 though the numerical simulation fails to simulate the secondary resistance peak under this condition. As shown in Section 2, this secondary resistance peak does not exist in other working condition experimental data. The error of the numerical simulation resistance increased from the low-speed segment of 6% to the high-speed segment of 33%. However, the numerical simulation resistance prediction under middle or low speed is more accurate. The trend of resistance is basically synchronous in the whole-speed segment. And the numerical simulation predicted the resistance peak at the Froude number of 0.46. It is also an acceptable accuracy compared with other side-wall hovercraft numerical simulation results.

It can be seen in Figure 18 that, the testing data of trim and sinkage is less than the corresponding simulation results. With increasing speed, in either trim or sinkage line graph, a positive agreement can still be observed between testing data and simulate data. The error of trim angle rages from 7%

to 15%, but actually, the value difference is merely 0.63°and 0.37°on average. On the other hand, the value difference is of heave is larger, while the trend also met a good agreement. It seems to be related that the hover system has a lower level air-flow loss in the numerical system. Therefore, the validity of the numerical method is approved. Furthermore, the relatively large error in the high-speed segment will a have a finite impact on the discussion of resistance reduction trend and effect in the following discussion.

**Table 4.** Numerical calculation and testing data comparison (Δ = 90 kg, Xgm = 150 mm, Q = 150 m³/h).

| V (m/s) | Fr | R/W | | a/(°) | | H/L | |
|---|---|---|---|---|---|---|---|
| | | CFD | Exp. | CFD | Exp. | CFD | Exp. |
| 0.5 | 0.09 | 0.005 | 0.003 | 1.37 | 1.28 | 0.0022 | 0.0025 |
| 1 | 0.18 | 0.016 | 0.015 | 1.65 | 1.43 | 0.0025 | 0.0021 |
| 1.5 | 0.28 | 0.030 | 0.038 | 1.80 | 1.55 | 0.0024 | 0.0011 |
| 2 | 0.37 | 0.055 | 0.073 | 2.21 | 1.68 | 0.0014 | 0.0002 |
| 2.5 | 0.46 | 0.079 | 0.093 | 2.79 | 2.24 | 0.0024 | 0.0004 |
| 2.8 | 0.52 | 0.069 | 0.081 | 3.12 | 2.71 | 0.0032 | 0.0030 |
| 3.2 | 0.59 | 0.079 | 0.091 | 3.49 | 3.13 | 0.0023 | 0.0021 |
| 3.6 | 0.66 | 0.084 | 0.106 | 3.67 | 3.04 | 0.0042 | 0.0011 |
| 4 | 0.74 | 0.096 | 0.123 | 3.54 | 3.01 | 0.0047 | 0.0022 |
| 4.5 | 0.83 | 0.111 | 0.139 | 3.07 | 2.82 | 0.0051 | 0.0033 |
| 5 | 0.92 | 0.130 | 0.196 | 3.15 | 2.78 | 0.0054 | 0.0042 |
| 5.5 | 1.01 | 0.139 | 0.184 | 3.03 | 2.69 | 0.0060 | 0.0050 |
| 6 | 1.11 | 0.156 | 0.203 | 2.98 | 2.59 | 0.0062 | 0.0056 |

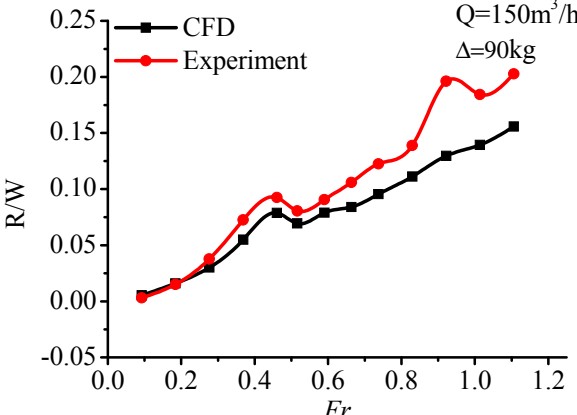

**Figure 17.** Resistance curves of CFD and experiment (Δ = 90 kg, Xgm = 150 mm, Q = 150 m³/h).

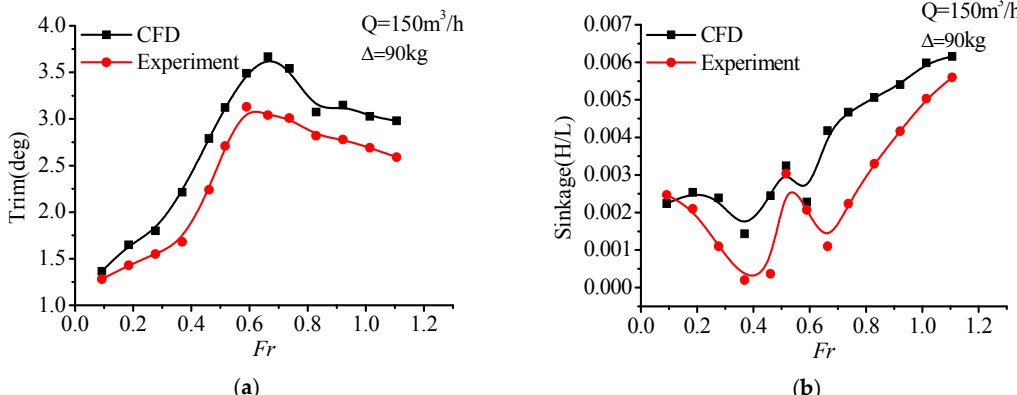

(a)

(b)

**Figure 18.** Motion attitude curves of CFD and experiment: (**a**) Trim angles curves; (**b**) Sinkage curves (Δ = 90 kg, Xgm = 150 mm, Q = 150 m³/h).

## 4. Discussion and Air Cushion Compartment

### 4.1. Air Cushion Compartment Model

Air Cushion compartment is a key technology in the upsizing of air cushion ships, which provides sufficient longitudinal and transverse stability for large hovercraft [14]. Similarly, the uniform pressure distribution in the hover air chamber can also be significant to the longitudinal and transverse stability of the PACSCAT while the complete consistency air cushion pressure at the bow and stern may not have an entirely favorable effect on the air cushion's navigational attitude. Eclectically, uniform distribution in the local region but with step-transmutation distribution, in general, a more favorable design criteria can be reached. Based on the air cushion compartment, regulating the pressure of each sub-cushions is also a means to improve the motion attitude.

From this section on, model-2 and model-3 are added through compartmentalization of the hover air chamber into two and three sub cushion, named model-2 (two cushions) and model-3 (three cushions), respectively. As shown in Figure 19, the cushion chamber volume is evenly divided with a cushion separator which is simplified as a rigid plate with a certain angle. The other geometric characteristics remain invariant. The bottom of the cushion separator is set to the same height with the stern air seal bottom. Especially, in model-3, the air total height of cushion separator near the stern is shortened to 85%. Thus, during the model motion process, the separator will not attach to the free surface when the trim angle increases, and the extra wave-making and friction resistance are avoided. The height is designed upon the difference between the height of the free surface at the maximum trim angle and chamber top panel.

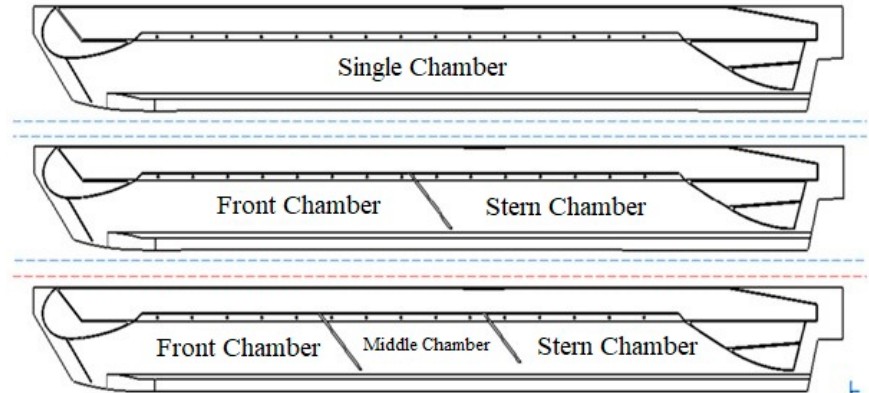

**Figure 19.** Models of air cushion compartment model-1 (single chamber), model-2 (two chambers) and model-3 (three chambers).

To measure the pressure distribution in the air hover chamber globally, the pressure monitoring points were added during the numerical simulation process, as shown in Figure 20. The monitoring points were arranged equidistantly from 1 to 8 along the air chamber. For the model-3, monitoring points 1 to 3 were located in the rear air chamber, monitoring points 4 and 5 were located in the middle chamber, and the rest 6 to 8 were located in the front.

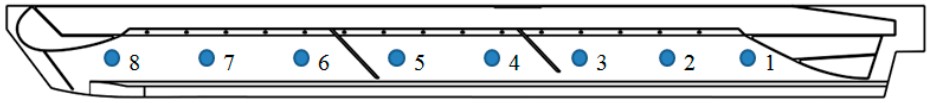

**Figure 20.** Distribution of cushion pressure monitoring points.

### 4.2. Relationship between Pressure Distribution and Air Cushion Compartment

Due to the process of simulation convergence, only the latter 60% part of the monitored air cushion pressure value (average of all points' acquisitions in a sub chamber) is extracted. The top panel of the

chamber is the main area the air cushion impacted on. Specifically, the global pressure value in the sub-chambers will determine the heave height. While the uneven pressure distribution will affect the trim angle stability at a constant speed motion until a new equilibrium pressure situation established.

The pressure value and distribution at any point of the cushion could be obtained from the results of the pressure plot of hover air chamber top panel (namely, area under the connecting bridge). By analyzing the regional air pressure distribution of condition ($\Delta$ = 90 kg, Q = 150 m$^3$/h) from Figure 21, the action mechanism of the air cushion compartment can be summarized as follows:

Self-comparison and cross-comparison of the air cushion compartment distributions of the single cushion, model-2, and model-3 were carried out. Where the self-comparison of these three models is taken with the increasing of speed individually, while the cross-comparison is taken commutatively with a constant speed.

In model-1, the pressure distributed uniformly in the hover air chamber. The range of pressure difference was not more than 30 Pa except for the low-pressure area around the intakes where the air is injected into the cushion which also proves the obvious pressure stabilizing effect of the plenum chamber. In addition, the pressure in the air cushion chamber fell at first and then increased to a peak with increasing speed (which *Fr* ranged from 0.37 to 0.52). Finally, it decreased to a smaller value. Simultaneously, the front chamber of model-2 and front and middle chambers of model-3 also showed similar pressure transformations. It shows there is a good agreement of the trend of pressure changing phenomenon and regularity between the simulation and testing data. And more details on self-comparison pressure distribution is analyzed in Section 3.2.

Through cross-comparison, it is obvious that the pressure of the stern chamber is greater than that of the front chamber in model-2. In model-3, the pressure value is successively decreased chamber by chamber from the front via the middle to the stern. The results show consistent agreement with that obtained in the tank drag test by the literature [15].

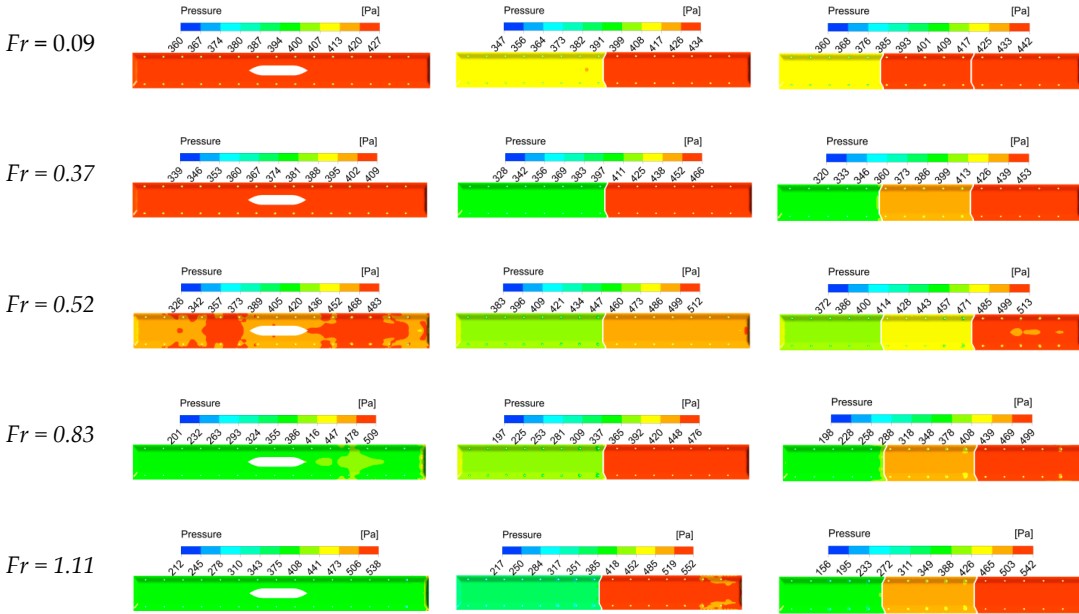

**Figure 21.** Pressure distribution of top panel of the hover chamber (model with single, two and three cushion from left to right).

For model-2, the absolute value of the stern chamber pressure also showed a fluctuation trend with increasing speed. During the Froude number ranging from 0.09 to 0.59, it climbed up at first, where after, it shows a downward trend until the Froude number reaches 1.01. The pressure ratio of front and stern chamber fluctuated at low-speed and resistance peak-speed segments and showed the disciplinary variation law at high speed. Numerically, the ratio increased with increasing speed and

reached a front stern pressure ratio of 0.69. In addition, in the figure, as for model-3, the air pressure of the front and stern chambers is fairly equal to that of the model-2. Furthermore, the pressure variation law and even the ratio value equal to each other. The middle air chamber is transitional, acting as a pressure buffer. The absolute pressure value still prevailed over that of the front chamber, which led to the model-3 having a better drag reduction effect than the model-2.

Figure 22 shows the dimensionless pressure value measured at point 1 of model-3 with increasing speed. It can be clearly seen that the absolute pressure value of the stern chamber is higher than that of the single cushion. Either the model-2 or model-2 does so which indicates that the air cushion compartment has a better effect on increasing the stern pressure. Especially during the high-speed segment, the pressure of the stern chamber of model-2 or model-2 is maintained at a high level, which provides favorable conditions for increasing the air pressure ratio between the front and stern chambers.

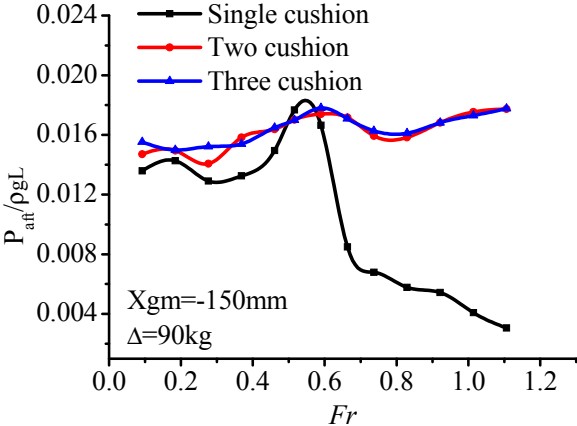

**Figure 22.** Comparison of dimensionless pressure value measured at point 1.

### 4.3. Relationship between Hydrodynamic Characteristics and Air Cushion Compartment

Analysis taken in Section 3.2 has proved that, in the low-speed segment ($Fr < 0.46$), with the increase in speed, the hull meets the characteristics of a conventional displacement ship. The resistance increased sharply in this speed period, but the cushion pressure played a limited role. Under a similar mechanism, as shown in Figure 23, the total resistance difference of the three models is slight at low speeds. At the speed of the resistance peak ($Fr = 0.46$), both model-2 and model-3 attain a distinct drag reduction effect which, compared to model-1, has the maximum drag reduction scope of model-2 and is about 15.28% in resistance peak segment, and the scope of model-3 is about 16.58%. After exceeding the hump, in a range of speed ($0.52 < Fr < 0.73$), the drag reduction effect of the modified models is not significant, both less than 7%. However, at the latter half of the high-speed segment, when the Froude number exceeds point 0.73, the effect becomes obvious again, and the reduction scale grows with increasing speed. At the Froude number of 1.11, the drag reduction scope of model-2 reaches 26.25%, and that of the model-3 reaches 35.8%.

Figure 24 shows the heave change trend with the increasing speed of the three models. The curves of modified models are up above model-1 all along. It means the heave value of the modified models, to some extent, are elevated. This trend is obvious in the whole-speed range. The heave, especially, reached the maximum increase amplitude at the resistance peak-speed segment. Figure 25 shows the trim angle changing trend with the increasing speed of the three models. It can be seen that the big difference of trim angle between model-1 and modified models exists at the speed section where the drag reduction effect is obvious. In the resistance peak-speed segment, the trim angle of the modified models is larger than that of the model-1, while the value relation is reversed during the latter half of the high-speed segment. Figure 26 shows the ratio between the front cushion pressure and the stern chamber calculated model by model with increasing speed. The ratio of the single cushion is 1, as the distribution uniformity pressure. The pressure ratios between the front cushion

and stern cushion of model-2 and model-3 equal to each other generally. And the pressure ratio trend is gradually decreasing, while inserted with an increasing trend at the resistance hump where the front pressure reaches its peak. The middle chamber of model-3 is a transitional chamber, so the ratio is also transitional.

Based on the analysis of total resistance, heave, trim angle, and pressure ratios above, it can be summarized, and the cushion compartment action mechanism can be explained as:

In the resistance peak-speed segment, both the front and stern chamber remain at a high pressure level. Which generate a high lift ratio, strengthens the hover effect, leading to a relatively obvious heave increase. On the other side, the pressure difference between the front and stern chamber is slight. The moment of pressure difference cannot restrain the uppitch, while the high air pressure of the front chamber makes the trim angle reach a large value in advance. Under the interaction of pressure difference moment and high lift ratio, cushion compartment technique improves the motion state and pressure distribute which result in, compared to model-1, the total resistance of modified models being reduced.

In the high-speed segment, the heave height and trim angle are both in a higher range. So is the bow seal height. Air leaking at the front chamber is severe, making it a low-pressure chamber. On the contrary, the stern chamber remains at a high level benefitting from the air cushion compartment. Moment of the front and stern chamber pressure difference act in the opposite direction from the hydrodynamic moment. Thus, the air cushion compartments can decrease the trim angle of this speed segment to achieve an optimal motion attitude and a better resistance reduction effect, as acting in low-speed segment.

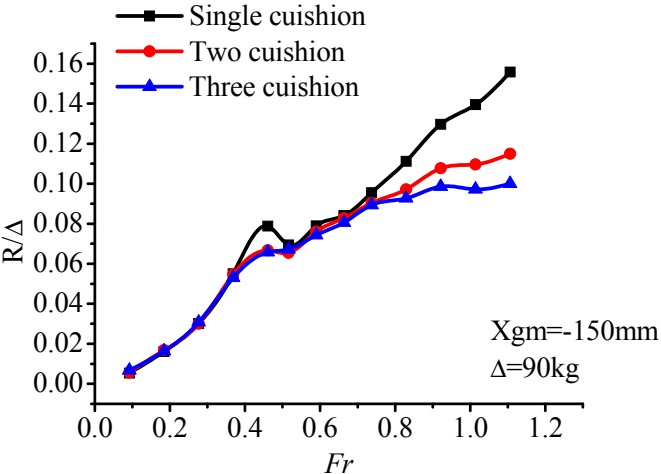

**Figure 23.** Comparison of models' resistance.

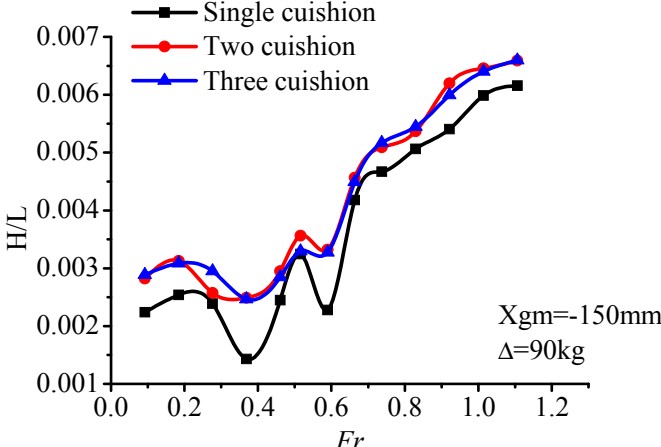

**Figure 24.** Comparison of models' sinkage.

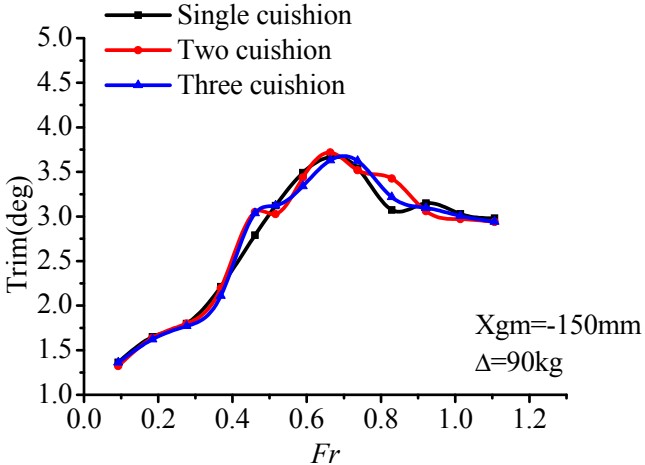

**Figure 25.** Comparison of models' trim angle.

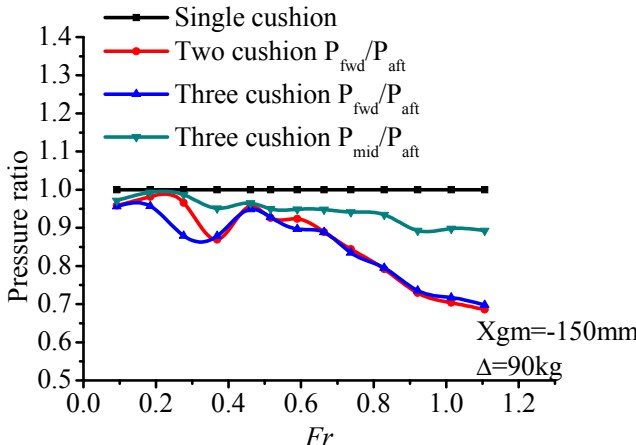

**Figure 26.** Comparison of pressure ratio between front and stern (middle) chamber.

## 5. Conclusions

In this study, experimental and numerical method were carried out for motion attitude and resistance characteristics. The resistance trial, resistance prediction method, and simulation of the three-dimensional flow field were launched, after that the influence of air hover system, such as the air cushion pressure distribution chamber compartment, on the hydrodynamic performance of the PACSCAT was analyzed. The main conclusions are as follows:

The wave making, motion attitude, and air cushion pressure are captured by the experimental scheme and the data acquisition method efficiently and directly. The resistance hump generally occurs around $Fr = 0.52$, and is slightly delayed toward a higher speed with increasing displacement. In the low-speed and high-speed segment, the influence of air cushion pressure on the PACSCAT motion attitude is limited, and the influence law is consistent with the hydrodynamic influence trend. However, during the resistance peak range, air cushion pressure changes the resistance peak value by affecting motion attitude.

A numerical simulation method with high accuracy was established. The numerical results of resistance and attitude were basically consistent with experimental results. Experimental phenomena were also reproduced in the numerical simulation. As mentioned above, the numerical results also showed similar rules between air cushion pressure and motion attitude. A slight impact at the low-speed segment while an obvious increasing at resistance peak and partial high-speed segment ($Fr = 0.46$–$0.83$) was observed.

Air flow loss was obvious in model-1, and by the method of air cushion compartmentalization, the stern (middle) chamber pressure could be maintained at a high level steadily. Air cushion compartmentalization achieved a superior drag reduction result by optimizing air cushion pressure distribution. At the high-speed range, compared with the original single chamber model-1, the drag reduction range of modified model-3 was about 22% on average, with a maximum of 35%. At the resistance peak-speed segment, the scope of drag reduction was about 11% on average, with a maximum of 16.58%. The results suggest that the air flow and pressure distribution have a significant influence on the hydrodynamic performance of the PACSCAT.

**Author Contributions:** Conceptualization, J.Z.; Formal analysis, S.L.; Investigation, S.L. and Y.Z.; Methodology, S.L., J.Z. and Z.G.; Resources, Y.Z.; Supervision, J.Z.; Validation, Z.G.; Writing–original draft, S.L.; Writing–review & editing, S.L. and J.Z.

**Funding:** The research was funded by National Science of Foundations grant numbers 51409069 and 51409054.

**Conflicts of Interest:** The authors declare no conflict of interest. The funders had no role in the design of the study; in the collection, analyses, or interpretation of data; or in the writing of the manuscript; or in the decision to publish the results.

## Appendix A

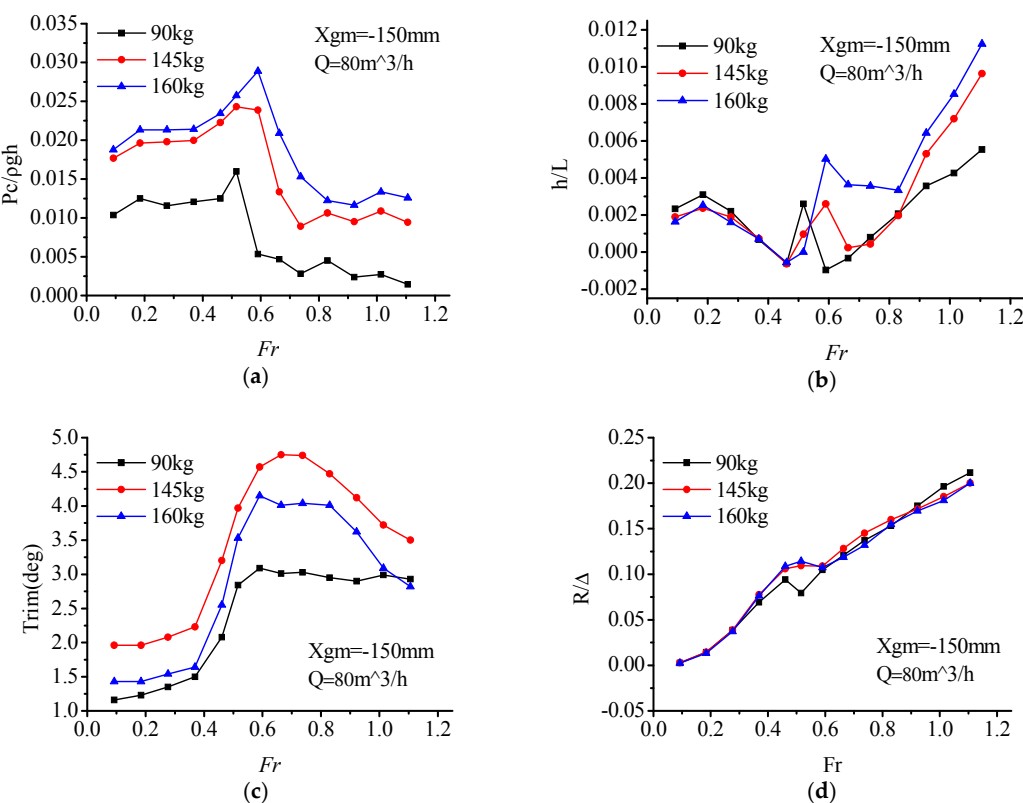

**Figure A1.** Comparisons of numerical and experimental results with consistent airflow Q = 80 m³/h. (**a**) Curves of air cushion pressure; (**b**) Curves of air cushion heave; (**c**) Curves of trim angle; (**d**) Curves of total resistance.

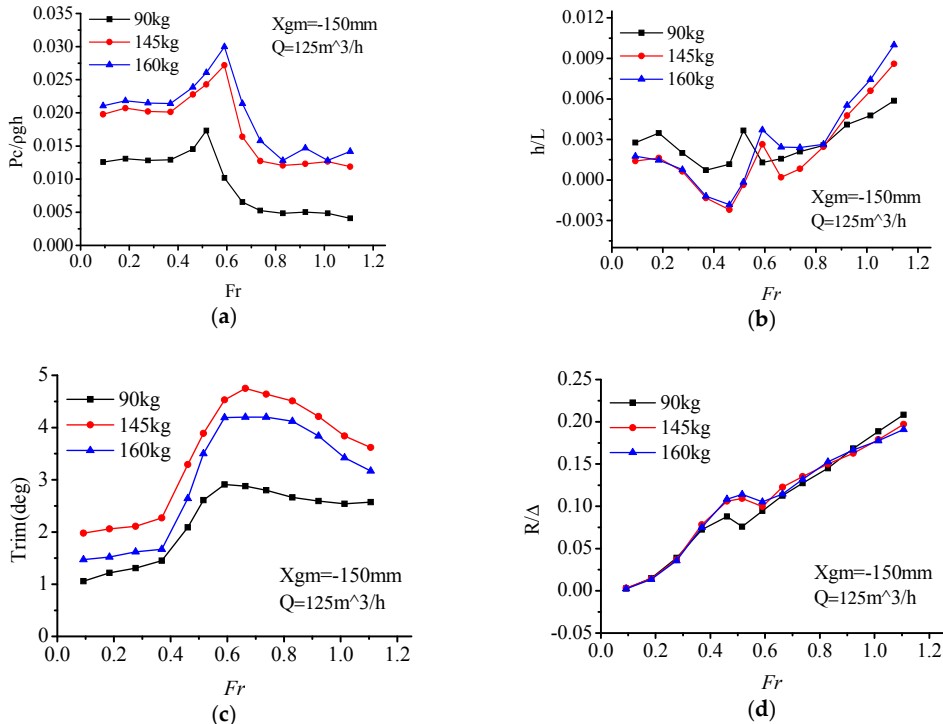

**Figure A2.** Comparisons of numerical and experimental results with consistent airflow Q = 125 m³/h. (**a**) Curves of air cushion pressure; (**b**) Curves of air cushion heave; (**c**) Curves of trim angle; (**d**) Curves of total resistance.

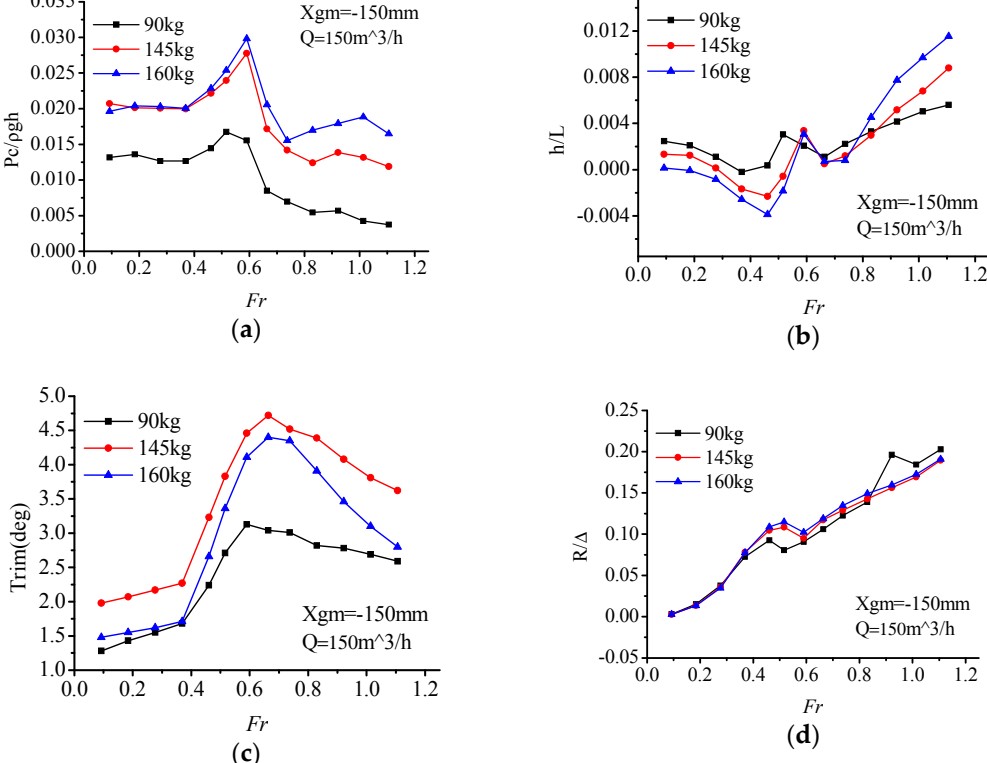

**Figure A3.** Comparisons of numerical and experimental results with consistent airflow Q = 150 m³/h. (**a**) Curves of air cushion pressure; (**b**) Curves of air cushion heave; (**c**) Curves of trim angle; (**d**) Curves of total resistance.

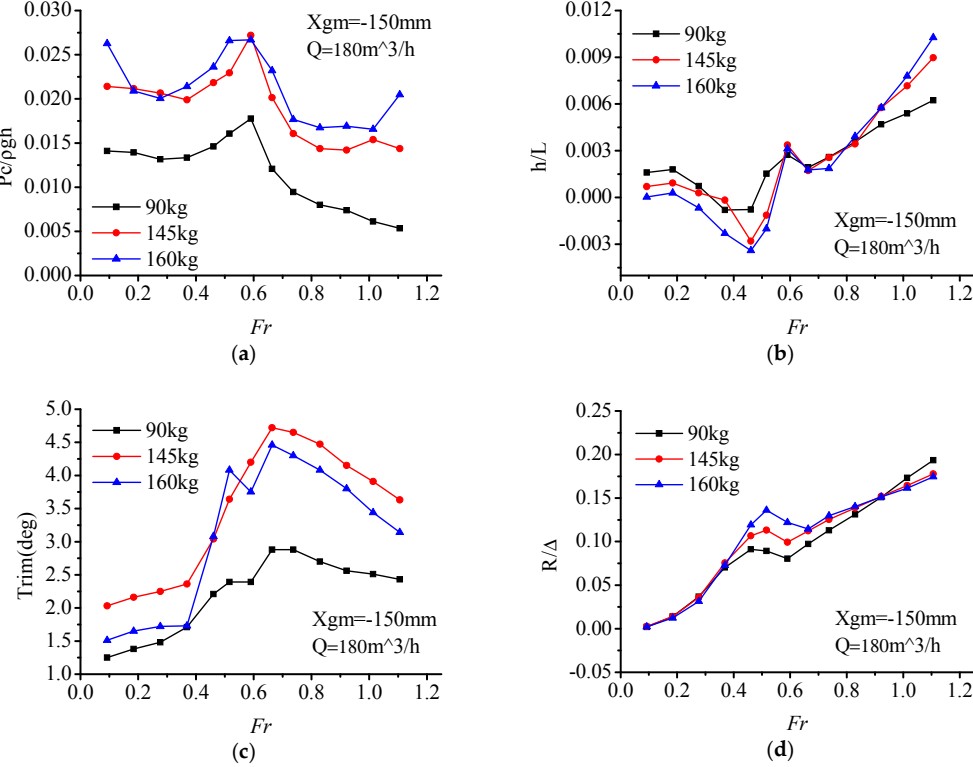

**Figure A4.** Comparisons of numerical and experimental results with consistent airflow Q = 180 m³/h. (**a**) Curves of air cushion pressure; (**b**) Curves of air cushion heave; (**c**) Curves of trim angle; (**d**) Curves of total resistance.

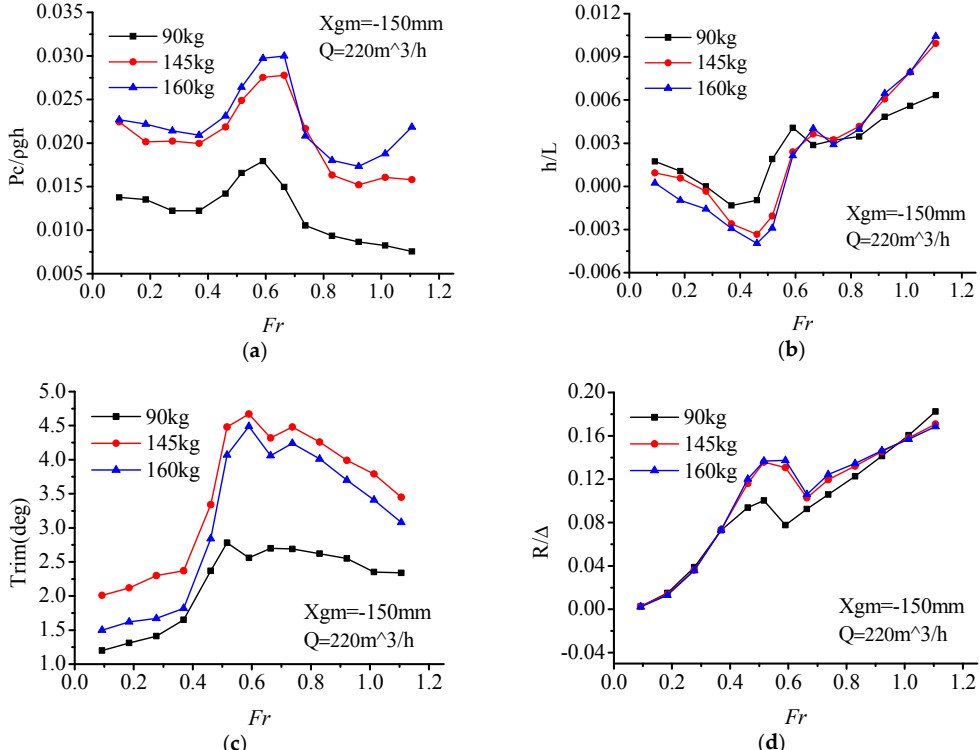

**Figure A5.** Comparisons of numerical and experimental results with consistent airflow Q = 220 m³/h. (**a**) Curves of air cushion pressure; (**b**) Curves of air cushion heave; (**c**) Curves of trim angle; (**d**) Curves of total resistance.

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
