# Peer review of "Experimental and Numerical Study on Motion and Resistance Characteristics of the Partial Air Cushion Supported Catamaran"

_water, doi:10.3390/w11051033_

Round 1

Reviewer 1 Report

This paper consider an Air Cushion Vehicle by experiment and numerical study. 

1: The paper has not been well organized. The reader confuses with the current structure.

2: English context needs to be revised. Several typo (angel instead of angle! , etc) and grammatical errors are found through the text.

3: Section 2.2 needs to be improved:

  3-1: It is worth providing picture(s) which show all your sensors and location of installation and inner gridline, and so on.

 3-2: How did you control the air pressure and airflow inside the cushion? You need to explain this part better.

 3-3: "The longitudinal position of the towing point is the same as that of the center of gravity, while the 91 vertical position should also be consistent with the center gravity as far as possible" what does this sentence mean?

4- Section 2.3: Why did you select parameters ( displacement 、 longitudinal center location and air flow rate ) for doing a series of experiments? What operational condition of a full scale ship is represented by each of which?

5: Numbering are repeated!

6: Sections 2.3 (mathematical...) and 2.4 are very brief and need to be revised for giving more complementary data of: how did you solve the equation, number of mesh and mesh resolution, a picture of full domain (2D and side view, preferably), etc.

7: In figures 5 to 7: it is not clear which data is for experiment and which is for simulation, captions and axis labels are not similar in some cases. 

8: Figure 10 and 11: domain which has been selected behind the ship (in experiment) is not enough to compare the results.

9: Give a definition  of parameters which are being considered in a table in its caption.

Author Response

Dear Editors and Reviewers:

Thank you for your letter and for the reviewers' comments concerning our manuscripten titled “Experimental and numerical study on motion and resistance characteristics of the partial air cushion supported catamaran (PACSCAT)” (ID:Water-466900). Those comments are all valuable and very helpful for revising and improving our paper, as well as the important guiding significance to our researches. We have studied comments carefully and have made correction which we hope meet with approval. Revised portion are marked inred in the paper. The main corrections in the paper and the responds to the reviewer's comments are as attached with the word file.

Thank you again for your kindly and constructive suggestions.

Best regards,

Reviewer 2 Report

This is a valuable paper which gives a new insight into the problem of the active control of PACSCAT (Partial Air Cushion Supported Catamaran). It is written in good English, still it requires some minor grammar corrections. Also, some sentences do not sound English, please try to correct stylistics.  

Reviewer has following remarks and recommendations:

- Authors consider the hull surface to be a free-slip moving boundary. Is it true?

- Authors use the SST turbulence model, but also state that use y+ of 60 for the hull surface grid. This does not correspond to the k-ω turbulence model. But in CFX, SST turbulence model uses the automatic near-wall treatment, which allows a smooth shift from low-Reynolds number form to a wall function formulation, according to the local velocity and mesh close to wall.

- Authors should provide details of the intakes, where air is injected into the cushion(s), as well as the appropriate boundary conditions used. Also, overall grid size should be published.

- The flow around the hull is unsteady, what is the time step used for simulations?

- In Figures 5-7, captions refer to the comparisons of numerical and experimental results. But graphs show just the experimental results. In Figure 7, please give the missing data for resistance curves.

- Text line 251 refers to Figure 9. Not correct.

- Figure 9: Trimaran or Catamaran?

- Table 3 and Figures 12, 13 correspond to Δ=90kg, Xgm=150mm, Q=150m3/h. Please specify clearly.

- Is there any comparison of data from 5 pressure sensor probes with 8 pressure monitoring points used during the numerical simulation process?

- Please, correct the reference 13 (text lines 504-505)

Author Response

Dear Reviewers:

Thank you for your comments concerning our manuscript entitled “Experimental and numerical study on motion and resistance characteristics of the partial air cushion supported catamaran (PACSCAT)” (ID: Water-466900). Those comments are all valuable and very helpful for revising and improving our paper, as well as the important guiding significance to our researches. We have studied comments carefully and have made correction which we hope meet with approval. Revised portion are marked in red in the paper. The main corrections in the paper and the responds to the reviewer's comments are attached with the word file below.

Thank you again for your kindly and constructive comments and suggestions.

Best regards, 

Reviewer 3 Report

Reviewer's comments on Manuscript ID: water-466900

Title: Experimental and numerical study on motion and resistance characteristics of the partial air cushion supported catamaran (PACSCAT)

1)      The introduction must provide a critical survey of the previous literature on the topic, and demonstrate a gap in the area of investigation. The introduction must be rewritten to be more inclusive.

2)      Provided equations for numerical simulation in section 2.3 are insufficient

3)      the result and discussion should be rewritten to provide extra explanatory rather than reporting the result

4)      Validation of numerical method must be introduced before the results and discussion

5)      Figure 5 -7 are presented as a comparisons of numerical and experimental results. However, from the figure experimental and numerical results cannot be distinguished.

6)      Nomenclature must be added to ease the understanding the text.

Author Response

(The authors gave the same response as above.)

Reviewer 4 Report

Topic very interesting.

Language is sometimes difficult to understand.

Typing errors.

During reading and analyzing a very interesting article and trying to understand the presented physical phenomena I found a set of unanswered questions.

Numerical model.

Numerical 2DOF model is correct. Boundary conditions correct. But....

line 151 -"The hull surface is considered as free-slip moving boundary....." Free-slip boundary conditions on the hull? Is this a clerical error or model error.

Inconsistent boundary conditions on the hull (y+ 60) with free-slip conditions.

How the flexible seals, bow and stern, with different geometry and different construction were modeled numerically?

The analysis of presented data would be easier if the same used parameters were defined, perhaps in graphical form.

It would be nice to have a sketch of motion parameters.

Fig. 5, 6, 7 as well A1---A5, do not contain of numerical simulations results so they can not present comparisons of experimental and numerical results.

The air supply conditions are not clear.  No seals details.  In another published article (Jinglei Yang, Zhuang Lin, Zeyang Gao and Ping L, A Study on the Motion of Rartial Air Cushion Support Catamaran in Regular Head Waves, Water , 20 March 2019) one can find more details of single air cushion system and details of seals.

The definition of h and H variables is required. Sketch with definitions would be useful.

The question is, how is possible to realize constant volume rate of supply air at variable pressure inside air chamber as is shown in Fig 5a, 6a, 7a?

Line 228 -"Further, hull wave-making resistance ........", then equation (6), then "Cf: frictional resistance coefficient". Where is the hull wave-making resistance?

Equation (4) contains other resistance components than shown in Fig. 8. Why is something presented and explained, then something else is depicted? And "othert" is not well defined?

Equation (8) uses a set of pressure parameters, dimensional parameters, even dynamic parameter - velocity. A sketch explaining definition of used names would be very helpful. If equation is taken from the literature, you can also download drawing with definitions.

Fig. 15 is useless, because there is no data in the work that represents pressure at these points.

Presentation of pressure distribution inside aircushion in form presented by authors (Fig. 16) looks nice, but it is useless. Due to the variable pressure range in each case, comparison is impossible. It may be more useful to present in the same sketch simply pressure distribution along the center line of the hull for single, double and triple chambers.

Line 359 " The range of pressure differences is no more 30Pa except the low pressure area around intakes where the air is injected into the cushion." OK, but where are injection sides? How is air injected? Is it a constant pressure source or constant volume rate source? How is it controlled in changing conditions? A change in pressure means what? Bad control of air supply system? Wrong operation of sealing curtains? Or something else?

Without information about air supply system, the analysis of pressure variation and pressure distribution is useless. What is the supply air volume rate distribution between chambers?

The strong point of the article is the presented physical interpretation of observed complex physical phenomena, including wave generation, high speed air flow leaks, water-pushing phenomena etc. Maybe some graphical presentation would be helpful.

Some typing errors.

The article is very interesting, useful for readers, but requires minor modifications.

The paper should be printed after explaining or clarifying the doubts and questions presented above.

Author Response

Dear Reviewer:

Thank you for your comments concerning our manuscript entitled “Experimental and numerical study on motion and resistance characteristics of the partial air cushion supported catamaran (PACSCAT)” (ID: Water-466900). Those comments are all valuable and very helpful for revising and improving our paper, as well as the important guiding significance to our researches. We have studied comments carefully and have made correction which we hope meet with approval. Revised portion are marked in red in the paper. The main corrections in the paper and the responds to the reviewer's comments are attached with the word file below.

Thank you again for your kindly and constructive comments and suggestions.

Best regards, 

Round 2

Reviewer 3 Report

the authors complied with the requested recommendation